# Microplastics and Per- and Polyfluoroalkyl Substances (PFAS) Analysis in Sea Turtles and Bottlenose Dolphins along Mississippi's Coast

Chanaka M. Navarathna [1,†], Hannah Pray [2,†], Prashan M. Rodrigo [1,†], Beatrice Arwenyo [1], Cassidy McNeely [1], Henry Reynolds [1], Natalie Hampton [3], Katherine Lape [1], Katie Roman [1], Maddie Heath [1], Sean Stokes [1], Sameera R. Gunatilake [4], Gombojav Ariunbold [5], Felio Perez [6], Rooban V. K. G. Thirumalai [7], EI Barbary Hassan [8], Islam Elsayed [8,9], Dinesh Mohan [10], Ashli Brown [11,12], Debra Moore [2,13], Stephen Reichley [2,13], Mark Lawrence [13,14] and Todd E. Mlsna [1,*]

[1]   Department of Chemistry, Mississippi State University, Starkville, MS 39762, USA
[2]   Department of Clinical Sciences, College of Veterinary Medicine, Mississippi State University, Starkville, MS 39762, USA
[3]   Department of Biological Sciences, Tougaloo College, Tougaloo, MS 39174, USA
[4]   College of Chemical Sciences, Institute of Chemistry Ceylon, Rajagiriya 10100, Sri Lanka
[5]   Department of Physics and Astronomy, Mississippi State University, Starkville, MS 39762, USA
[6]   Material Science Lab, Integrated Microscopy Center, University of Memphis, Memphis, TN 38152, USA
[7]   Institute of Imaging and Analytic Technology (I2AT), Mississippi State University, Starkville, MS 39762, USA
[8]   Department of Sustainable Bioproducts, Mississippi State University, Box 9820, Starkville, MS 39762, USA
[9]   Department of Chemistry, Faculty of Science, Damietta University, New Damietta 34517, Egypt
[10]  School of Environmental Sciences, Jawaharlal Nehru University, New Delhi 110067, India
[11]  Department of Biochemistry, Mississippi State University, Starkville, MS 39762, USA
[12]  Mississippi State Chemical Laboratory, Mississippi State University, Starkville, MS 39762, USA
[13]  Global Center for Aquatic Health and Food Security, Mississippi State University, Starkville, MS 39762, USA
[14]  Comparative Biomedical Sciences, College of Veterinary Medicine, Mississippi State University, Starkville, MS 39762, USA
*    Correspondence: tmlsna@chemistry.msstate.edu; Tel.:+1-662-325-6744; Fax: +1-662-325-1618
†    These authors contributed equally to this work.

**Abstract:** Global plastic production and usage has increased annually for decades and microplastic pollutants ($\leq$5 mm) are a growing concern. Microplastics in surface waters can adsorb and desorb harmful chemicals such as per- and polyfluoroalkyl substances (PFAS). Microplastics can accumulate across all tropic levels in the marine food web. The purpose of this research was to analyze the stomach and intestinal contents of stranded (Mississippi coast) bottlenose dolphins and sea turtles for the presence of microplastics and commonly found PFAS, PFOS, PFOA, and GenX. Gut contents were digested (10% KOH in 50% MeOH) and then analyzed for microplastics using pyrolysis gas chromatography-mass spectrometry (Pyro-GC-MS), Nile red microscopy, X-ray photo electron spectroscopy (XPS), and Raman spectroscopy. Digested sample filtrate was pre-concentrated using solid-phase extraction (SPE) before PFAS liquid chromatography-tandem mass spectrometry (LC-MS/MS) analysis. The PFOS extraction and analysis had 98.6% recovery when validated with certified pike-perch fish reference material. The Nile red testing on most samples revealed the presence of microplastics (Table S1). The Pyro-GC-MS results from two samples confirmed the presence of the plasticizer acetamide. The Raman spectroscopy analysis indicated characteristic plastic peaks corresponding to polystyrene in one sample. PFOS (95.5 to 1,934.5 µg/kg) was detected in three dolphin stomach samples. This project is part of a long-term study with the goal of a better understanding of microplastics and PFAS environmental contamination and their impact on bottlenose dolphins and sea turtles.

**Keywords:** microplastics; dolphin; sea turtles; PFAS; PFOS; Pyro-GC-MS; analysis

## 1. Introduction

Plastic production is estimated to double over the next 20 years and almost quadruple by 2050 worldwide [1]. At present, it is estimated that 13 million tons of plastics pollute the water each year [2]. The pollution may be linked to unsuccessful recycling, littering, and even weather events that cause plastics to travel [3]. Plastic has become popular due to its wide variability in usage. Plastic materials have allowed for energy conservation, lower costs, mass production, and a variety of advances in society [4]. However, with the qualities that cause plastic to be a desirable material, the harsh reality of its non-biodegradable properties arises [5]. This effect is the accumulation of plastics and microplastics that will last hundreds of years in the world's oceans. It is estimated that 50–80% of the pollution that occurs within the oceans is due to plastics [6]. Although the properties do not allow the complete disintegration of plastics over a short period, larger fragments can be broken down into smaller microplastics based on their distinctive physical and chemical characteristics and the environment [7].

The microplastic size on its longest side can be defined as <5 mm [8]. The category can be further broken down into many shapes, including spheres, fibers, pellets, film, and irregular fragments [9]. Primary microplastics originate from plastic-producing industries, facial wash, soap, cosmetics additives, and waste. In contrast, secondary microplastics may originate from clothing materials [9] or have been broken off into larger pieces by the mechanical and chemical stresses that fabrics undergo during a washing process [7].

Microplastics are hazardous due to their small nature and their resemblance to food causing animals to ingest them unknowingly or willingly [10]. Microplastics also have a high surface area to volume ratio that allows for the potential of heavy metal and organic pollutant absorption [11]. Fish often ingest microplastics containing toxic chemicals. Once absorbed in the body, these toxins can cause several irreversible problems. Even without chemicals, the physical properties of microplastics can block the intestinal tract and overcrowd the stomach [11]. This then leads to a lack of appetite in fish, which can ultimately result in death due to starvation. These fish can be eaten by predators, which some believe leads to a trophic transfer of microplastics [11,12]. The continuation of the microplastic throughout different species is dangerous, as there is no way to filter them out of the oceans due to the size of the particles [3]. Microplastics are slipping their way into the environment, affecting not only ecosystems and animals but also, in the long run, humans [13].

As microplastics are an emerging area of study, scientists are finding increasing amounts in various samples. Microplastics have been studied in the environment, with scientists looking closely at sediments [11] and seawater [11] where organisms live to better understand plastics' prevalence. Smaller organisms, plankton [14], sea vase [15], shrimp [16], and crabs [17] have also been tested for plastic contamination. Fish from around the world [7,17–26] are being used to learn how different fish are affected by microplastics and if they have any trace of particulate matter. Larger animals, such as dolphins [18], sea turtles [19–21], and whales [8,22], have also been investigated for microplastics.

MPs could conceal further dangers, such as their association with persistent organic pollutants (POPs) such as PFAS, dioxane, pesticides, chloroethanes, etc. [23]. When POPs are exposed repeatedly, they can accumulate in the environment in high quantities and eventually leak into living things, affecting the function of their organs.

PFAS are a group of manmade chemicals [24]. They were first introduced in 1940 because they have excellent chemical and thermal resistant, hydrophobic, zwitterionic, and surface-active properties and hence are used in a wide variety of applications in the US and around the globe, including non-stick, stain-resistant coatings, microwaveable food packaging, firefighting foams, cosmetics, waterproof clothing, electronics, etc. [25]. Common PFASs are perfluorinated carboxylic acids (PFCAs), perfluorocarbon sulfonic acids (PFSAs), perfluorooctanoic acid (PFOA), and perfluoro-octane sulfonate (PFOS). These compounds are highly persistent in the environment because of the stable C-F backbone.

The high electronegativity of F-atoms causes C-F bond to be very short and strong. The C-F bond has bond energy between 485 kJ/mol and 585 kJ/mol [26]. At concentrations as low as 70 ng/L levels, they cause deleterious effects, including development effects, cell cycle alteration [3], infertility, and carcinogenicity [27–29]. Upon exposure to even low ppm levels, they can bioaccumulate in the body for up to 9 years because of their high fat solubility [27–29]. Alarmingly high PFAS concentrations have been found in various bottlenose dolphin organs/tissues. For example, Charleston bottlenose dolphin plasma contained 49-1,171 ng/g PFOS [30]. Up to 62,973 ng/g PFOS were detected in the livers of bottlenose dolphins stranded along the northern Adriatic Sea [31]. The total PFAS in 12 liver samples from New Zealand ranged from 11.3 to 110.4 ng/g (median = 34.1) [32].

The meticulous extraction of MPs without altering their chemistry or morphology by optimizing the digestion process by utilizing different extraction solvents and co-extraction PFAS without losing the recovery has not yet been the subject of any research. This is crucial to speed up many analyses, accurately analyze MPs, and track them to their source. Hence, in this work, the stomach contents of bottlenose dolphins and Kemp's ridley sea turtles obtained from The Institute of Marine Mammal Studies (IMMS) in Gulfport, MS, were used. The dolphins in the Northern Gulf of Mexico are top predators. The Kemp's ridley sea turtle is considered the world's most critically endangered. The dietary contents of dolphins sampled include a variety of digested fish and shrimp. At the same time, the Kemp's ridley sea turtle gut contained crabs, other crustaceans, and primarily fish (from opportunistic feeding). The objectives of this study were to (i) optimize a complete digestion method without altering the plastic morphology and chemical texture, (ii) discover if microplastics existed in the samples, and (iii) evaluate their characteristics such as size, color, shape, and type of plastic, and (iv) to develop and optimize a method to pre-concentrate and analyze targeted PFAS (PFOS, PFOA, and GenX) in the stomach contents in both dolphin and sea turtles.

## 2. Materials and Methods

All the plastic standards and chemicals were purchased from Sigma-Aldrich unless otherwise noted.

### 2.1. Digestion Optimization

To the best of our knowledge, no previous work has reported the systematic development of plastic extraction protocols from stomach contents. Many studies have employed different strengths of KOH, HCl, and $H_2O_2$ primarily to digest animal tissues but they have paid little or no attention to the chemical interactions that can occur with polymers and cause their breakdown.

Careful plastic extraction without altering their chemistry and morphology is crucial. Hence, we systematically investigated sample digestion and plastic extraction. For this purpose, several standard plastics (polyethylene [PE], polystyrene [PS], polyvinyl alcohol [PVA], polyvinyl chloride [PVC], polyamide [PA], polyurethane [PU], polypropylene [PP], and polyethylene terephthalate [PET]) were immersed in different digestion solutions (% HCl, 4% HCl, 6% HCl, 8% HCl, 10% HCl, 5% KOH, 10% KOH, 15% KOH, 20% KOH, 25% KOH, 5% $H_2O_2$, 10% $H_2O_2$, 15% $H_2O_2$, 20% $H_2O_2$, 25% $H_2O_2$, 30% $H_2O_2$, 5% KOH in 50% MeOH, 10% KOH in 50% MeOH, 15% KOH in 50% MeOH, 20% KOH in 50% MeOH, and 25% KOH in 50% MeOH) that are commonly used to digest animal tissues [33]. Briefly, a 100 mg portion of each plastic was added into a test tube for each solution and 10 mL of the solution was added in a 1:100 $w/v$ ratio. After 24 h on the shaker, the mixtures were vacuum filtrated using a glass funnel and dried on a watch glass. The Fourier-transformed infra-red (FT-IR) spectra for each plastic before and after digestion were recorded and compared. The peak shifts and the emergence of new peaks were indicators of the plastic chemistry or morphological changes.

### 2.2. Digestion of Tissues

The stomach and intestine samples for MP and PFAS analysis were excised from each carcass using a stainless-steel knife and placed in aluminum bags before storage at −20 °C. The samples were thawed at 2–8 °C then allowed to warm to room temperature (25–30 °C). The gastrointestinal content of the dolphin and sea turtle stomachs and intestines were labeled and each large sample was divided into 10 smaller sections. The smaller sections were weighed to 2.00 g. A 200 mL of 10% KOH in 50 % MeOH solution was added to each due to the 1:100 $w/v$ ratio. The samples were placed on a stir plate at room temperature until the entire content was completely digested for approximately 48 h. Next, the solution was vacuum filtrated (5 μm pore size and 47 mm cellulose nitrate filter) to retrieve the microplastics. The filter paper would dry overnight on a watch glass before the examination.

### 2.3. Characterization of Microplastics

Characterizing MPs is a challenging task. Several analysis approaches must be used depending on the size of the MP (Figure 1). Therefore, we combined several analytical methods for the identification, including simple visual inspection, vibration spectroscopy, mass spectroscopy, and X-ray photoelectron spectroscopy [34].

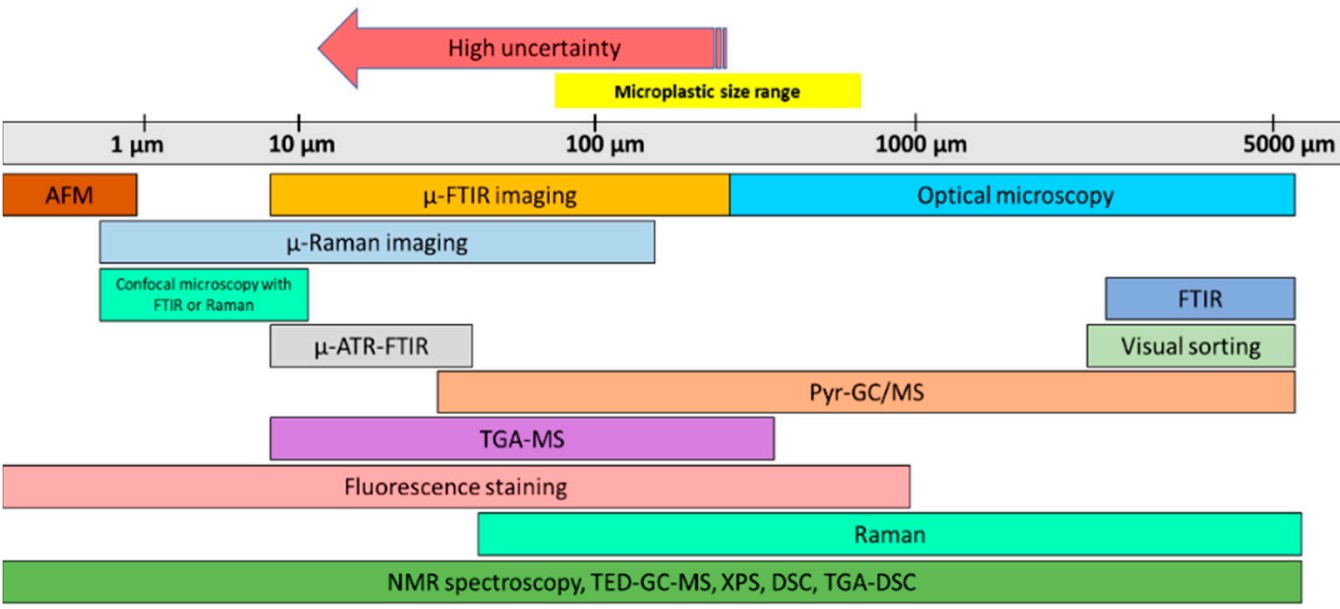

**Figure 1.** Summary of MP analysis techniques. "Adapted with permission from Ref. [34]. 2020, CRC Press.

Nile red (NR) is a hydrophobic, metachromatic, and photochemically stable dye. Its staining is a simple-to-use approach for analyzing a wide spectrum of microplastics [35]. Nile red was (10 mg/L in acetone) sprayed (6–8 times) on each filter paper until the contents were completely covered. The samples were observed under a handheld UV light (365 nm). The fluoresced particles were counted along with their fluorescence color and shape. The functional groups were examined using FT-IR for the larger particles and then compared to standards. To further classify the plastics, a density separation tube was used to find the density range of larger plastic fragments (Figure 2). The solutions used in the separation were methanol, Milli-Q water, NaCl, ZnCl$_2$, and Na$_6$O$_{39}$W$_{12}$.

### 2.4. Plastic Confirmation

Pyrolysis gas chromatography mass spectrometry (Pyro-GC-MS), X-ray photo electron spectroscopy (XPS), and Raman spectroscopy were used to identify plastic-like particles that are challenging to identify using microscopy. The pyro-GC-MS oven was held at

40 °C for 4 min and then ramped to 280 °C at a rate of 5 °C/min. The injection temperature was 280 °C, split ratio = 5:1, carrier gas = He, transfer temp = 223 °C, source temp = 250 °C, and the mass scan was performed in the 30–550 Da range using a 30.0 m × 320 µm column. The X-ray photoelectron spectroscopy (XPS) measurements were conducted with a Thermo Scientific K-Alpha XPS system equipped with a monochromatic X-ray source at 1486.6 eV, corresponding to the Al $K_\alpha$ line, with a spot size of 400 µm. The photoelectrons were collected from 90° takeoff angles relative to the overall sample's fractal particle surface and measured in the constant analyzer energy mode. Low resolution survey (LR) and high resolution (HR) core level spectra were extracted at 200 eV and 40 eV pass energies, respectively. The XPS data were deconvoluted and analyzed using "Avantage v5.932" software. The Raman spectroscopy was performed with an initial beam, with a center wavelength of 1035 nm, produced from an ytterbium-doped amplified fs fiber laser (MXR-Clark) with an average power of 10 W at a 1 MHz repetition rate.

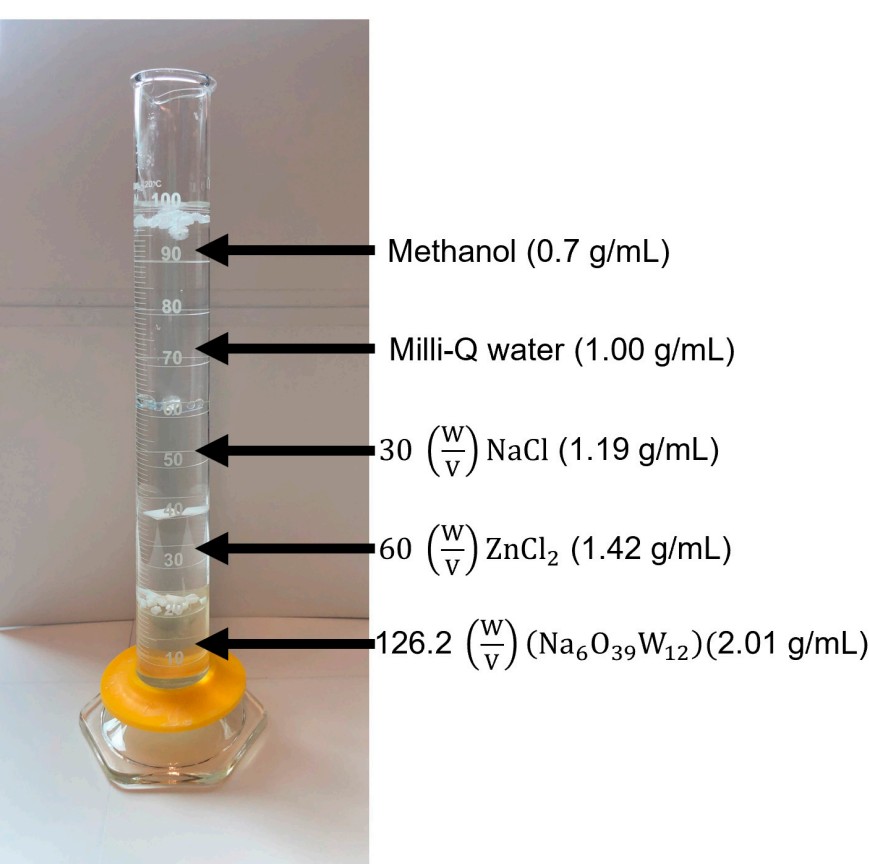

**Figure 2.** An example density separation tube with polystyrene, polyurethane, polyethylene terephthalate, and polypropylene.

*2.5. PFAS Analysis*

The digests from the plastic extraction were heated at 70 °C for 2 h to boil off the methanol. The aqueous solution was passed through weak anionic exchange [polystyrene divinylbenzene (PSDVB)] solid phase extraction (SPE) WAX (diamino) cartridges. Then, the PFAS were stripped with methanol 10 mL. Then, the final volume was reduced to 2 mL using a nitrogen evaporator and transferred to polypropylene liquid chromatography vials. The EPA method 573.1 was used with some modifications employing a Dionex Ultimate 3000 U-HPLC equipped with a Bruker microTOF-QII ESI mass spectrometer [36]. Ammonium acetate (5 mM) (A) and ammonium acetate (5 mM) in 95 % methanol (B) were used in the mobile phase. An Agilent InfinityLab Poroshell 120 EC-C18, 2.1 × 100 mm, 2.7 µm, narrow bore C18 LC column was used. A 10% B flow held for 0.5 min, linearly increased to 30% within 2 min, 95% within 14 min, and then held until 14.5 min (and 6 min

post time) as the program gradient was used. A 0.4 µL/min flow was used with an injection volume of 25 µL. The negative ion mode was used for mass detection. The extracted ion chromatograms were recorded (PFOS—498.9 m/Z and PFOA—368.9 m/Z) and the peaks were integrated using HyStar software to construct the calibration curves. The method's detection limit (LOD) was ~7 ng/L and ~10 ng/L for PFOS and PFOA, respectively. The IRRM-427 pike-perch fish certified reference material (CRM) (PFOS = 16.0 ± 1.7 ng/g) was used to validate the digestion, extraction, pre-concentration, and analysis method.

*2.6. Quality Control*

To ensure that no airborne plastic contamination occurred, blank filter papers were exposed to the open air in the laboratory while filtering digestive fluids. The presence of microplastics was evaluated by first filtering 1 L of tap water following the same process as with the samples and the procedural blanks. Next, KOH solutions were prepared with Milli-Q water and filtered before the addition of chromatography-grade methanol for sample digestion. The samples were handled and kept in glass only. Throughout the procedure, single-use gowns were worn on top of white cotton laboratory coats and blue nitrile gloves, following health and safety regulations. The ATR diamond and its base were thoroughly cleaned with acetone before and after the procedure, between every sample, and between measurements of the same sample. Prior to every sample scan, the spectrometer scanned the background 8 times. Fluoropolymer-free LC-MS/MS plumbing (PEEK), PFAS-free chromatography solvents, and a delay column were used in the PFAS analysis to prevent contamination. Scheme 1 shows a summary of the workflow for MP and PFAS extraction and analysis.

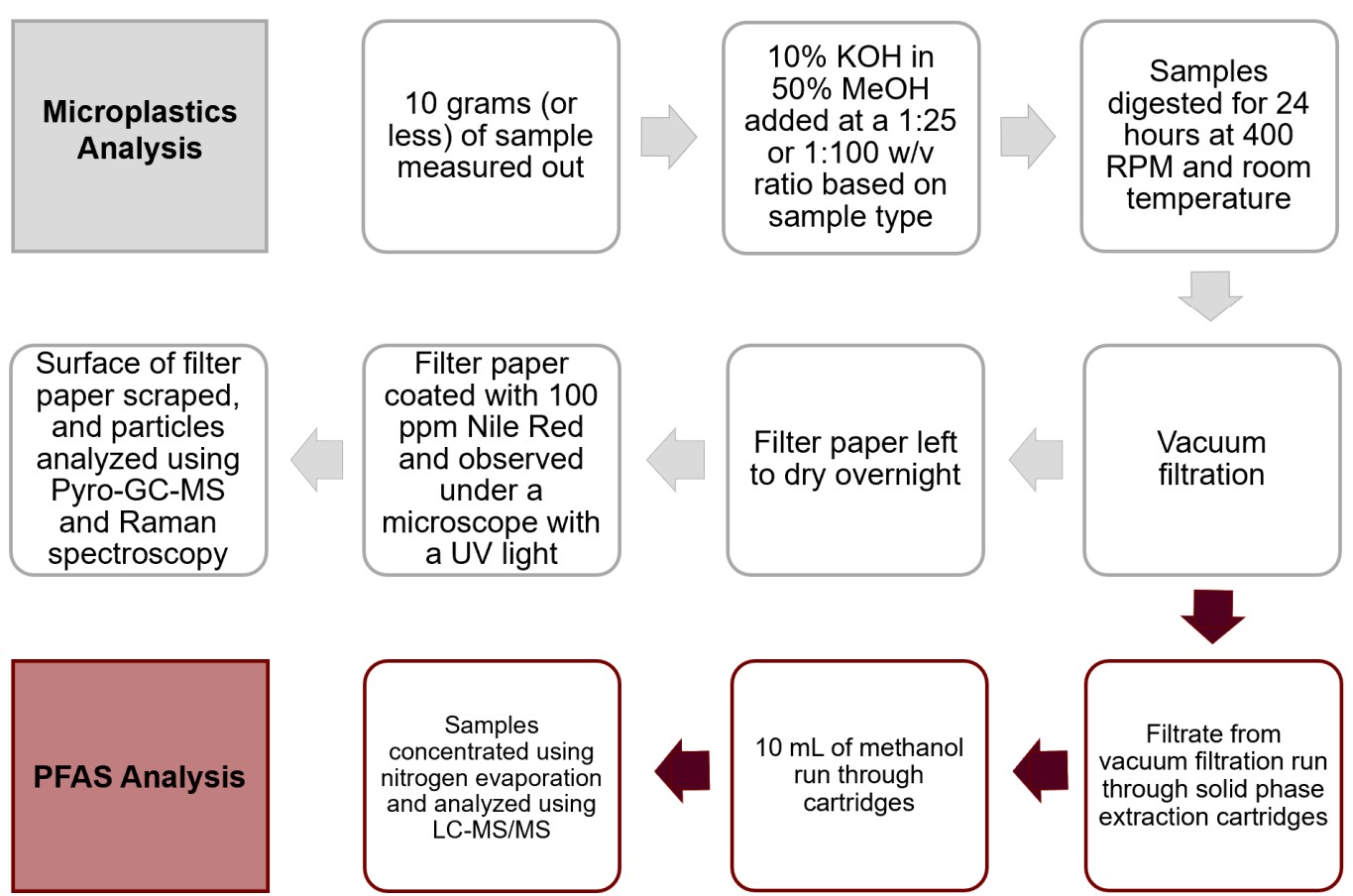

**Scheme 1.** Summary of the workflow.

### 3. Results

*3.1. Digestion Optimization*

Figures 3 and 4 display the FTIR spectra obtained for standard plastics and the spectra obtained after mixing them with the digestion solutions. The criteria for selecting the digestion mixture were the absence of new peaks or no significant peak shifts; 10% KOH in 50% MeOH was chosen for digesting the tissues as no or minimal spectral changes were observed compared to the neat IR spectra. A new broad peak developed in the 3300–3600 cm$^{-1}$ region after PA, PET, PU, PVA, and PVC (versus their standards) (Figures 3 and 4) were exposed to 5–30% $H_2O_2$, HCl, and KOH. This may be because amide or ester groups were hydrolyzed to carboxylic −OH and amine (−$NH_2$) moieties. Broadness increased with increasing solution strength from 5% to 30% indicating further hydrolysis, hence the degradation of plastic polymers and revealing more −OH groups. PE, PP, and PS did not display this change due to the absence of hydrolyzable groups.

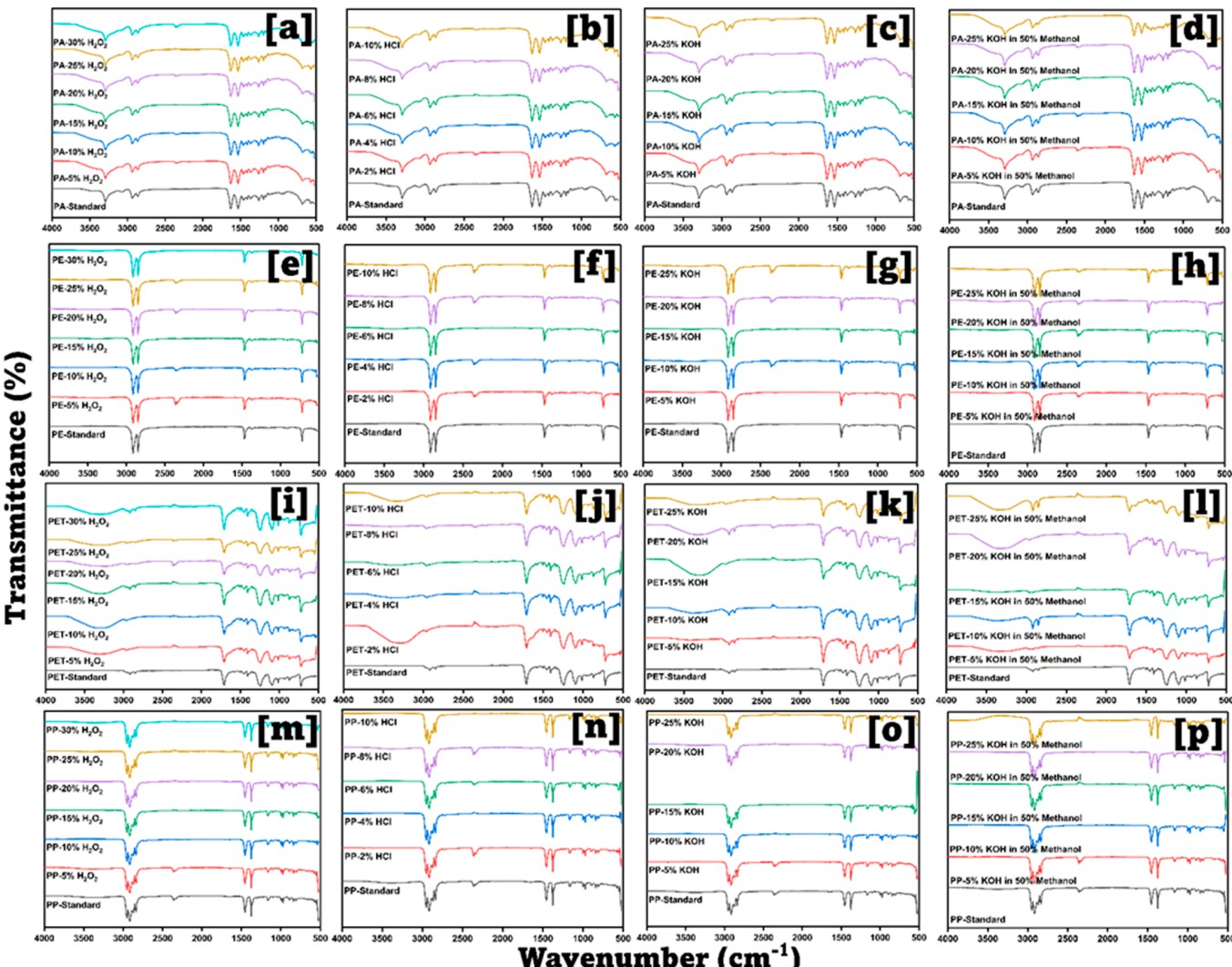

**Figure 3.** FT−IR spectra for polyamide, polyethylene, polyethylene terephthalate, and polypropylene in each concentration of KOH, HCl, $H_2O_2$, and KOH in MeOH.

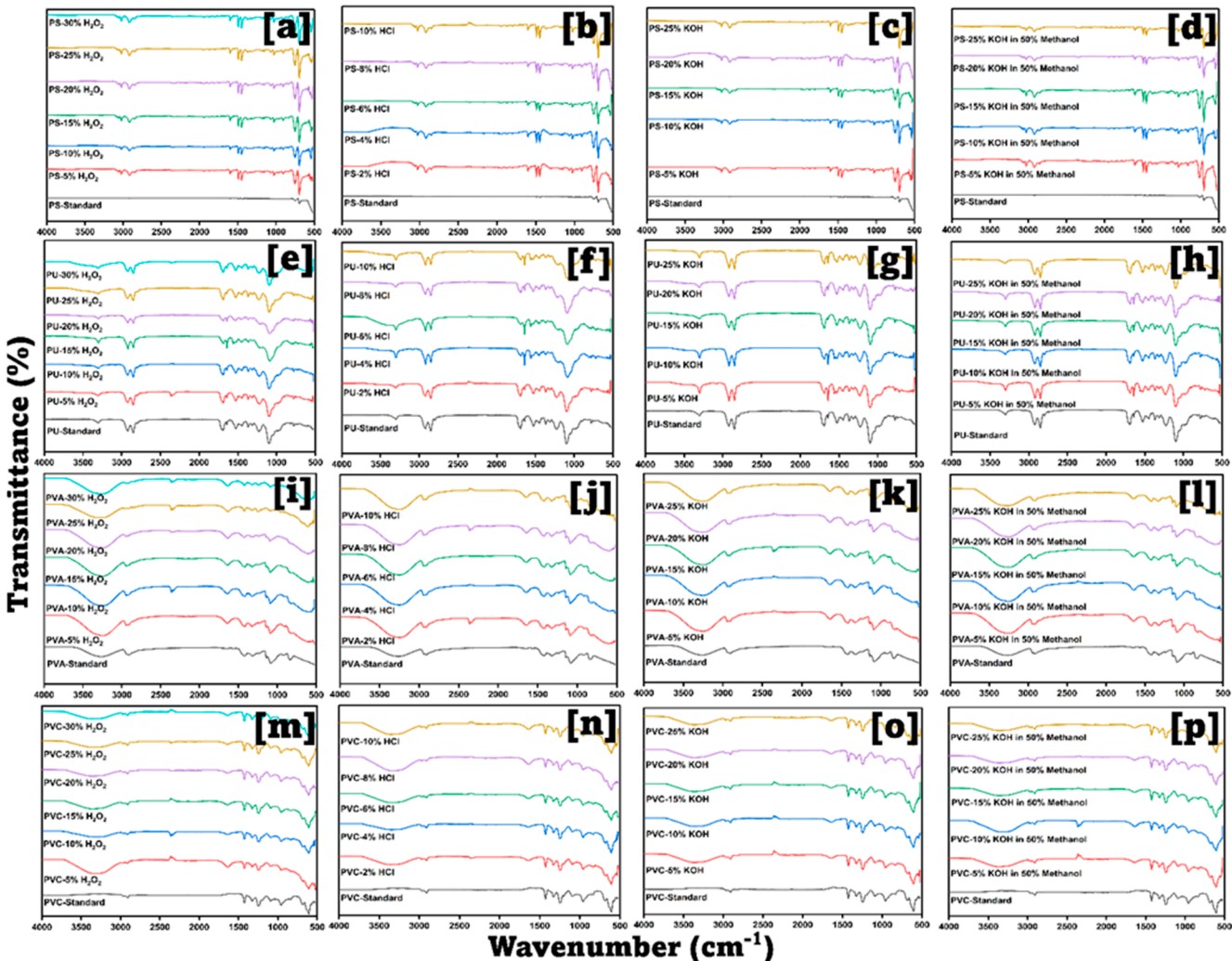

**Figure 4.** FT−IR spectra for polystyrene, polyurethane, polyvinyl alcohol, and polyvinyl chloride in each concentration of KOH, HCl, H$_2$O$_2$, and KOH in MeOH.

The 10% KOH in the 50% MeOH mixture appears relatively inert for plastics and selective for the tissue material. Several studies reported using 10% KOH for plastic extraction from dolphin tissues [37,38]. Our digesting process aimed to extract both MPs and PFAS simultaneously. This combination was chosen because PFAS are highly soluble in methanol [39]. Furthermore, this mixture showed very low turbidity, which suggests that the digestion is complete.

### 3.2. Characterization of Microplastics

Two large fragments of plastics were found in the sea turtle's intestinal sample. Their images and characteristics are provided below (Figure 5).

Low-density PE has a density range of 0.91–0.94 g/mL and high-density PE ranges from 0.93 to 0.97 g/mL [5]. Since the transparent plastic piece floated on the water layer but sunk in the methanol, the density is consistent with that of polyethylene. The colored plastic floated on the NaCl layer (1.00–1.42 g/mL), which matched the density of PVA, is 1.19 g/mL [40]. The FT-IR spectra for both plastics were compared with the standard PE and PVA spectra and matched with minor differences. This could be due to potential plastic degradation by the stomach enzymes, which could add or remove original functional groups, thereby altering IR adsorption [41]. Further, the XPS survey spectra elemental percentages suggest that PE and PVA are the predominant plastics.

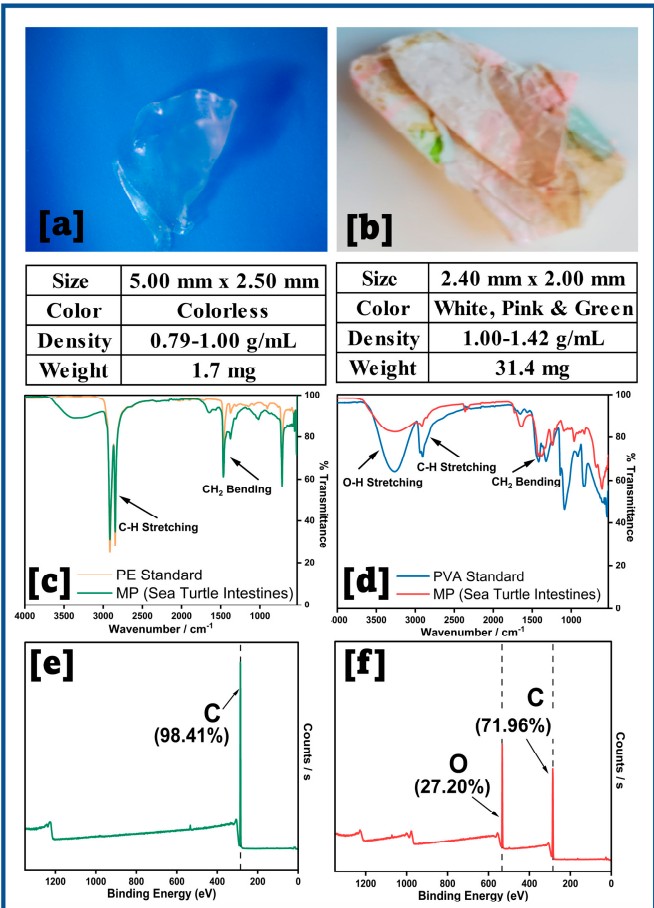

**Figure 5.** Large MPs extracted from (**a**,**b**) sea turtle intestines, FT–IR spectra for (**c**) PE/colorless, (**d**) PVA/colored plastic, and (**e**,**f**) Low resolution survey XPS spectra.

The Nile red positive microplastic results were found in the majority of tissue samples (Figure 6). In some of the samples, the microplastics were aggregated and caused it to be difficult to numerically differentiate, which is noted by "TMTC" (Too Many To Count) (See supporting material Table S1). The color that is noted is the color in which the microplastics fluoresced. Nile red allows microplastics to be easily visualized in the samples; however, it will only fluoresce PE, PP, PS, PC, PUR, and PEVA [35]. Thus, more microplastics that do not fluoresce may be present than those seen by the UV light alone. In addition, many samples had too many microplastics to define numerically; thus, when leading to the total per sample, a ">" is used to denote the possibility of more than is visibly seen. Nile red testing may be prone to false positives due to biofilm coatings on undigested bones and trace tissues [42]. No accurate characteristics besides number and fluorescence color could be retrieved for the smaller microplastics (Figure 6). They were too small to measure the length and weight when placed in the density separation tube or with IR.

The pyrolysis-GC-MS results for two dolphin stomach contents depict a match based on mass spectrums between acetamide at a retention time of 7.307 min (Figure 7a) and pyrrole at 6.787 min (Figure 7b) for some residues scrapped from the filter papers. These two chemicals are widely used as plasticizers, suggesting the potential presence of microplastics in the tested samples [43]. The ideal pyrolysis conditions for pyro-GC/MS research should also be established because the thermally degraded plastic products vary depending on the pyrolysis temperature. The data processing stage requires time because pyro-GC/MS results are currently manually interpreted. Therefore, polymer identification and quantification automation is a viable area for further research [44].

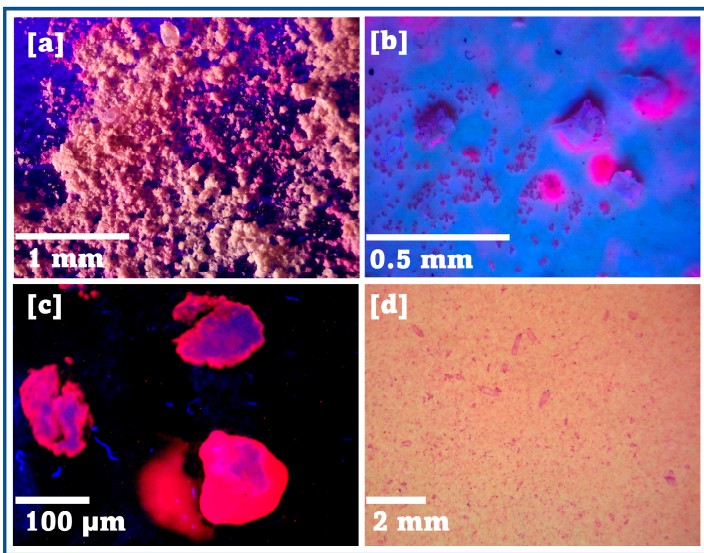

**Figure 6.** (**a**–**d**) Example images of Nile red-stained MPs to fluoresce under UV light.

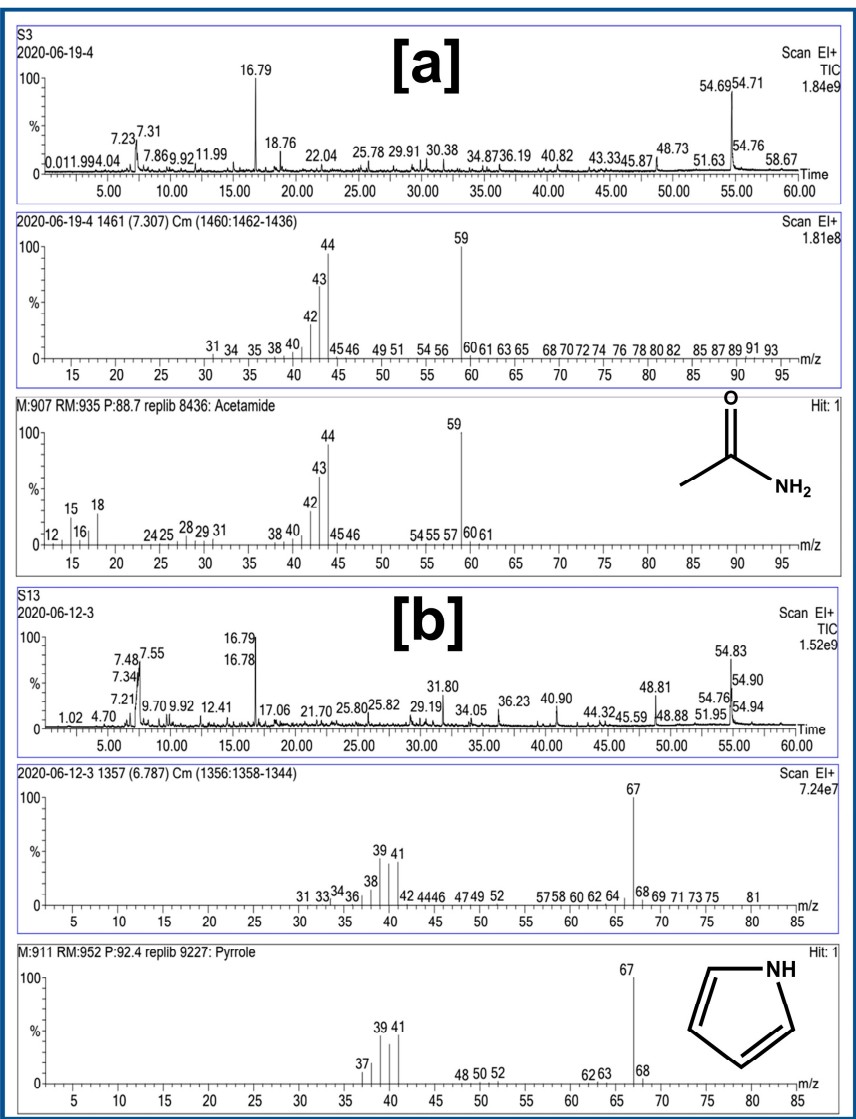

**Figure 7.** Pyro-GC data for selected plastic samples (**a**) S3 (acetamide) and (**b**) S13 (pyrrole).

A selected plastic sample was subjected to Raman spectroscopy analysis. The sample contained peaks that matched PS and PMA (Figure 8). The extent of the biofilm coating [42] on the plastic and particle size broadened the Raman signal, causing it to be challenging to match standards.

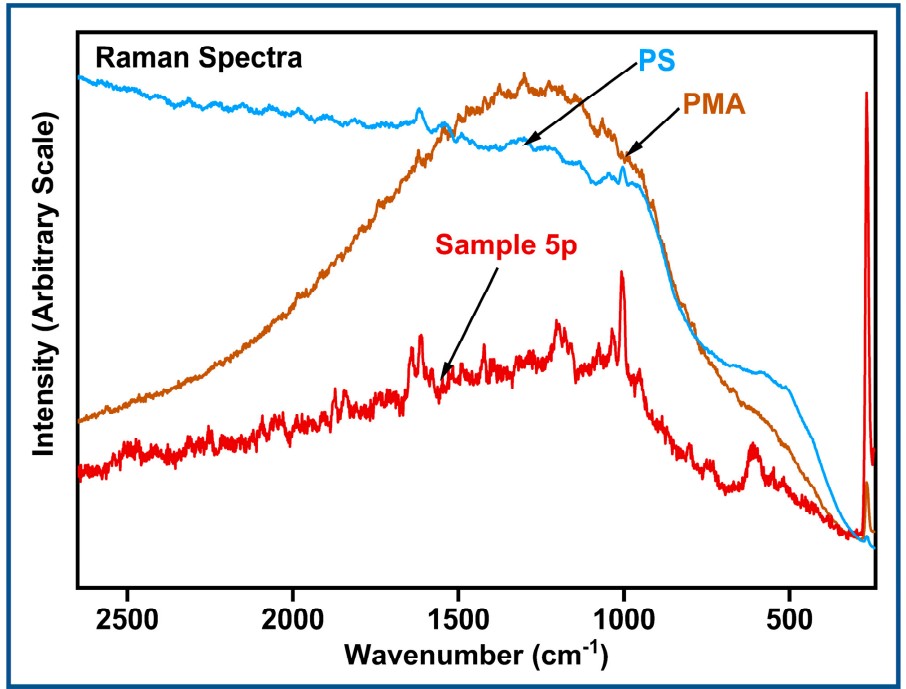

**Figure 8.** Raman spectroscopy data for a selected plastic sample.

Due to the limited number of samples available, no statistical interpretation was performed. However, a summary of plastics detected is presented in Table 1. It should be noted that the physical characteristics of the animals (such as age, sex, body size, and reason for death), as well as various microplastic extraction techniques, sample organs, feeding patterns, and habitats, may all significantly contribute to this diversity in microplastic abundance [38].

**Table 1.** The total amount of microplastics per sample type.

| Tissue Sample (Number Analyzed) | Total Number of Microplastics (Nile Red) | Number of Large Plastics | Positive for PFOS | Confirmed by Pryo-GC |
|---|---|---|---|---|
| Dolphin Stomach Contents (10) | >104 | 0 | 3 | 2 |
| Sea Turtle Stomach Contents (10) | >60 | 2 | 0 | 0 |
| Dolphin Intestinal Contents (10) | >134 | 0 | 0 | 0 |
| Sea Turtle Intestinal Contents (10) | >136 | 0 | 0 | 0 |

### 3.3. PFAS Analysis

For an analytical procedure to be employed, the recovery percentages should typically be in the 85–115% range. The pike-perch CRM provided 98.6% recovery for PFOS, indicating that the PFOS extraction method is satisfactory and analytically acceptable. The PFOS, PFOA, and GenX contents were below the detection limits (7–10 ng/L) of

most samples tested. Only three dolphin samples were positive for the presence of PFOS (Figure 9) and they were in the from 95 to 1934.6 µg/kg range. High exposure levels of PFASs in Indo-Pacific humpback dolphins (liver content) from the PRE, China, were observed [45]. The source of the PFAS in these animals is unknown. However, microplastics are well known for the vector transport of hydrophobic organic contaminants, including PFAS [46].

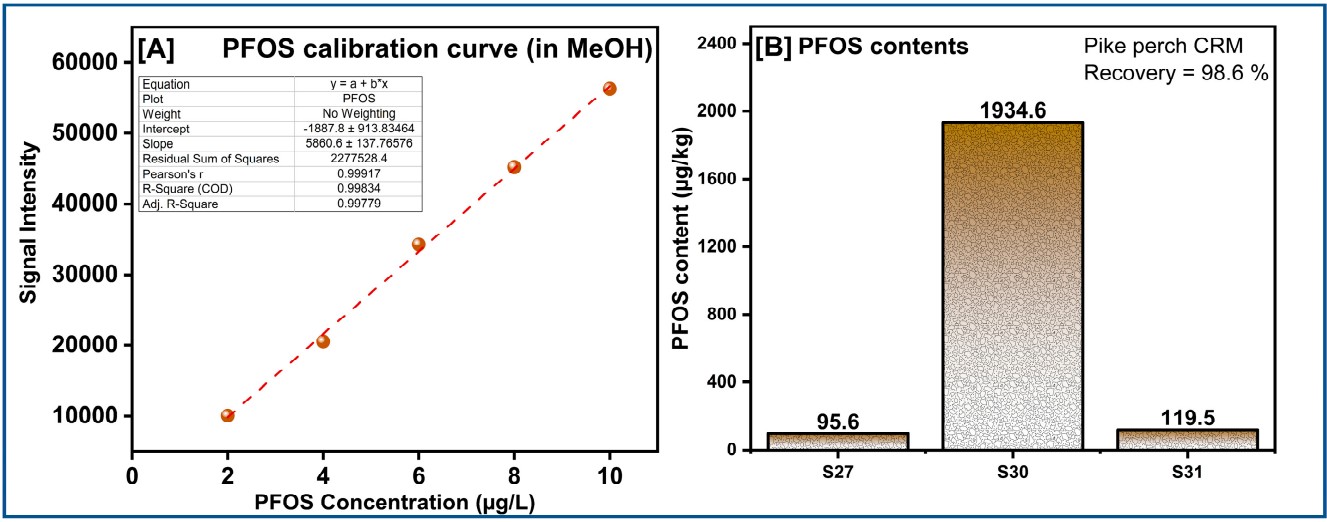

**Figure 9.** (**A**) PFOS calibration curve in MeOH and (**B**) PFOS contents detected in dolphin stomach contents.

## 4. Conclusions

An efficient and robust analytical method was developed, optimized, and validated to co-extract and analyze MPs and PFAS from dolphin, sea turtle stomachs, and intestines. This study documented the first description of stomach and intestine MPs and PFOS in bottlenose dolphins and sea turtles from the Mississippi Gulf coast. We also presented a systematic workflow to identify microplastics using various analytical techniques. According to our preliminary findings, bottlenose dolphins, ordinary coastal residents who spend the entire year in the gulf, could be used as a bioindicator to track microplastic contamination. Future research is necessary to properly understand the health effects of microplastics on this endangered species of cetacean, given the high rates of microplastic ingestion and the high microplastics found in these animals. These findings illustrate that animals at the top of the food chain are consuming plastics or microplastics either directly or indirectly from lower trophic level species. It would also be essential to identify chemicals that microplastics absorb and bioaccumulate in the body and understand the organs or body systems affected. Future research will concentrate more on statistically examining the amount of plastic dependent on animal age, sex, weight, and other bodily sections (liver and kidney tissues, etc.). Additionally of interest will be the adsorption of PFAS, pesticides, and other emerging contaminants onto microplastics and both targeted and nontargeted analysis of a broad spectrum of PFAS (long chain, short chain, and oxygenated).

**Supplementary Materials:** The following supporting information can be downloaded at: https://www.mdpi.com/article/10.3390/analytica4010003/s1, Table S1: The characteristics of microplastics found in each sample, where DS stands for dolphin stomach, DI stands for dolphin intestine, STS stands for sea turtle stomach, and STI stands for sea turtle intestine.

**Author Contributions:** Conceptualization, C.M.N., P.M.R., S.R.G. and T.E.M.; methodology, C.M.N and P.M.R.; software, C.M.N., G.A. and E.B.H.; validation, D.M. (Debra Moore); formal analysis, F.P., G.A., E.B.H. and I.E.; investigation, C.M.N., H.P., P.M.R., B.A., C.M., H.R., N.H., K.L., K.R. and M.H.; resources, S.S. and R.V.K.G.T.; writing—original draft preparation, C.M.N. and C.M.; writing—review and editing, C.M.N. and C.M.; visualization, D.M (Dinesh Mohan); supervision, A.B., D.M. (Debra Moore), S.R., M.L. and T.E.M.; project administration, A.B., D.M., S.R., M.L. and T.E.M.; funding acquisition, A.B., D.M. (Debra Moore), S.R., M.L. and T.E.M. All authors have read and agreed to the published version of the manuscript.

**Funding:** This research was funded by Mississippi Department of Marine Resources (Gulf of Mexico Energy Security Act) and National Institute of Health 5T35OD010432.

**Data Availability Statement:** The data presented in this study are available on request from the corresponding author.

**Conflicts of Interest:** The authors declare no conflict of interest.

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
