# Peer review of "Microplastics and Per- and Polyfluoroalkyl Substances (PFAS) Analysis in Sea Turtles and Bottlenose Dolphins along Mississippi’s Coast"

_analytica, doi:10.3390/analytica4010003_

Round 1

Reviewer 1 Report

The present article addresses a topic of great importance nowadays: the implications of microplastics in animals Health, more specifically in sea turtles and bottlenose dolphins.  The structure and the methods that authors used are suitable. However, I have some appointments.

·         Most of the images need improvement, as FTIR images and others

·         In case of the FTIR perheaps is better show only one example and the others put in supplementary material. And it is important to identify the characteristic peaks of each plastic.

·       Add a  reference in figure 1

·       Figure 6, the figure don´t have scale

·      The pictures seem to be very concentrated, because don´t have good contrast between microplastics and filter. Perheaps is better reduce/decrease the quantity of Nile-red and ensure the filter is dry. ( see the https://doi.org/10.1016/j.scitotenv.2019.07.060 and https://doi.org/10.3390/polym14142902)

·       Improve the Pyro GC chromatograms

·    In conclusion, it would be interesting to show which MP was detected and try to relate the contaminants with the source. 

Author Response

The present article addresses a topic of great importance nowadays: the implications of microplastics in animals Health, more specifically in sea turtles and bottlenose dolphins.  The structure and the methods that authors used are suitable. However, I have some appointments.

Response - Thank you for your comments and suggestions to improve the manuscript

  • Most of the images need improvement, as FTIR images and others

Response - Figures were improved in the revised version of the manuscript as suggested

  • In case of the FTIR perheaps is better show only one example and the others put in supplementary material. And it is important to identify the characteristic peaks of each plastic.

Response – We have improved the FTIR figures and the discussion as well including key IR peaks into the discussion

  •      Add a  reference in figure 1

Response - We have added a reference for this figure

  •      Figure 6, the figure don´t have scale

Response – Scale bars were included

  •     The pictures seem to be very concentrated, because don´t have good contrast between microplastics and filter. Perheaps is better reduce/decrease the quantity of Nile-red and ensure the filter is dry. ( see the https://doi.org/10.1016/j.scitotenv.2019.07.060 and https://doi.org/10.3390/polym14142902)

We appreciate your advice. With lower Nile Red concentrations (1-5 mg/L), we had trouble seeing these tiny particles. The article that was recommended to read dealt with microplastic fibers, which were reasonably noticeable at low concentrations. Also, different plastics has different degree of interaction with Nile Red and we wanted to use a universal concentration that would allow us to visualize all the different MPs with a single Nile Red spray. In our future publications, we'll further optimize the Nile Red concentrations more systematically.

  •      Improve the Pyro GC chromatograms

Response – GC chromatograms were improved

  •   In conclusion, it would be interesting to show which MP was detected and try to relate the contaminants with the source. 

Only two of the huge plastic fragments we discovered could have been identified. This is only a small portion of a long term study, and we are now working with many samples to understand distributions, various plastic types, sizes, and shapes, among other things. In our upcoming publication, we'll pay closer attention to pinpointing the plastic's true origins. The current study has mostly concentrated on the improvement of digestion and the presentation of preliminary results for selected samples.

Author Response

Thank you for your comments and suggestion to improve our manuscript. We have provided a rigorous response to your comments below.

  1. We have included the following statement before beginning the last paragraph of the introduction to justify the research gap and the importance of this work.

“The meticulous extraction of MPs without altering their chemistry or morphology by optimizing the digestion process by utilizing different extraction solvents and co-extraction PFAS without losing the recovery has not yet been the subject of any research. This is crucial to speed up many analyses, accurately analyze MPs, and track them to their source.”

Also, the following statement was added to the digestion optimization section to justify the choice of H2O2, KOH, and HCl chemicals in this study.

“Many studies have employed different strengths of KOH, HCl, and H2O2 primarily to digest animal tissues, but they have paid little or no attention to the chemical interactions that can occur with polymers and cause their breakdown.”

Although the digestion optimization is one focus of this work, we also have presented the MP and PFAS data for the bottlenose dolphin and sea turtles found in the MS Gulf coast for the first time, and this was adequately discussed in the introduction and elsewhere

  1. We have added the following discussion on cartridge type in the PFAS analysis section

Polymeric polystyrene divinylbenzene (PSDVB) sorbent

WAX chemistry—diamino functionality

pKa > 8

Please refer to the following link for more information

https://lcms.cz/labrulez-bucket-strapi-h3hsga3/brochure_bond_elut_pfas_wax_spe_cartridge_5994_4996en_agilent_63005e1db4/brochure-bond-elut-pfas-wax-spe-cartridge-5994-4996en-agilent.pdf

We have not used isotopically labeled PFAS for our testing as 13C labeled standards are very expensive and were not affordable due to funding constraints. However, we have validated our analytical method using a pike-perch fish certified reference material sample subjected to similar digestion. The PFAS recovery % was following satisfactory levels.

  1. We have improved the discussion on choosing 10% KOH in 50% methanol solution for our digestion. The newly added writings are highlighted in the revised version
  2. The figures were improved as suggested by the reviewer
  3. O1s and Na1s may have occurred due to trace contamination levels and water adsorption onto the plastic surface. Those assignments were removed in the revised version for clarity.
  4. The manuscript was proofread to correct typos and any language issues.